# Structure-Directing Interplay between Tetrel and Halogen Bonding in Co-Crystal of Lead(II) Diethyldithiocarbamate with Tetraiodoethylene

**DOI:** 10.3390/ijms231911870

**Published:** 2022-10-06

**Authors:** Lev E. Zelenkov, Daniil M. Ivanov, Ilya A. Tyumentsev, Yulia A. Izotova, Vadim Yu. Kukushkin, Nadezhda A. Bokach

**Affiliations:** 1Institute of Chemistry, Saint Petersburg State University, Universitetskaya Nab. 7/9, 199034 Saint Petersburg, Russia; 2School of Physics and Engineering, ITMO University, 191002 Saint Petersburg, Russia; 3A. E. Favorsky Institute of Chemistry, Siberian Branch of the Russian Academy of Sciences, 664033 Irkutsk, Russia; 4Institute of Chemistry and Pharmaceutical Technologies, Altai State University, 656049 Barnaul, Russia

**Keywords:** tetrel bonding, halogen bonding, non-covalent interactions, lead(II) dithicarbamates, QTAIM, NCI

## Abstract

The co-crystallization of the lead(II) complex [Pb(S_2_CNEt_2_)_2_] with tetraiodoethylene (C_2_I_4_) gave the co-crystal, [Pb(S_2_CNEt_2_)_2_]∙½C_2_I_4_, whose X-ray structure exhibits only a small change of the crystal parameters than those in the parent [Pb(S_2_CNEt_2_)_2_]. The supramolecular organization of the co-crystal is largely determined by an interplay between Pb⋯S tetrel bonding (TeB) and I⋯S halogen bonding (HaB) with comparable contributions from these non-covalent contacts; the TeBs observed in the parent complex, [Pb(S_2_CNEt_2_)_2_], remain unchanged in the co-crystal. An analysis of the theoretical calculation data, performed for the crystal and cluster models of [Pb(S_2_CNEt_2_)_2_]∙½C_2_I_4_, revealed the non-covalent nature of the Pb⋯S TeB (−5.41 and −7.78 kcal/mol) and I⋯S HaB (−7.26 and −11.37 kcal/mol) interactions and indicate that in the co-crystal these non-covalent forces are similar in energy.

## 1. Introduction

Non-covalent interactions [1,2,3,4,5,6] (NCIs) play a key role in the modulation of supramolecular organizations [7,8,9,10,11,12,13,14], which is useful for targeted crystal design [7,8,9,11,12,15,16,17,18,19] and the fabrication of functional materials [20], as well as for enhancing catalytic activity [21,22] and reactivity [23,24,25,26,27,28]. All of these factors contributed to the motivation behind supramolecular chemistry studies utilizing a diversity of NCIs. The understanding of non-covalent forces is uneven with regards to the placement of the donor atoms in the periodic table. In this context, hydrogen- [29,30], halogen- [31,32,33], and chalcogen [34] bonds were widely studied (and obtained their IUPAC names [30,33,34]) in contrast to other types of non-covalent forces [4,7,35], such as tetrel bonding.

Tetrel bonding (abbreviated as TeB) belongs to the group of interactions that include Group 14 donors (namely, C/Si/Ge/Sn/Pb), whose understanding and applications are very far from exhaustive. Several reviews [35,36] considered the known examples of TeBs that occurred between a tetrel atom (functioning as a σ-hole donor) and a lone-pair possessing atom (acting as a nucleophilic component of the TeB linkage). Most of the recent reports on TeB have been focused on contacts involving carbon atom(s) [35,37,38,39] or silicon atom(s) [40,41], while TeB including σ-hole-donating tin [42] or lead sites [36,43,44,45,46,47] are still poorly investigated. Notably, before our decade, the ability of lead(II) to act as a Lewis acid in intermolecular contacts was considered in the frameworks of additional coordination contacts, secondary bonding, or semi-coordination [36,48].

The coexistence and combined effects of two or more types of NCIs are important for the rational design of solid materials. Combinations of hydrogen and halogen bonds [49,50,51,52,53,54,55], hydrogen and chalcogen bonds [56,57], and halogen and chalcogen bonds [25,58] in one system and their interplay have been verified, while the combined effects of TeB together with any other kind of NCI are, so far, little explored. In particular, several papers [59,60,61,62] outlined the coexistence of Pb···S(N) TeBs with π-stacking and/or hydrogen bonding in solid crystals. An interplay between TeB and hydrogen bonding was studied theoretically for the XCN/4-EF_3_-pyridine (X = Cl, Br; E = C, Si, Ge) [63] and NH_3_/EF_3_X (E = C, Si, Ge, Sn; X = Cl, Br, I) systems [64].

Our recent reports verified various approaches for the supramolecular assembly of transition metal dithiocarbamates and -carbonates with halogen bond (abbreviated as HaB) donors [65,66,67]. In pursuit of that project, we turned to lead(II) species, where a positively-charged Pb^II^ site could function as a σ-hole-donating component of TeB [35]. The Cambridge Structural Database (CSD) search on the supramolecular organization of various homoleptic lead(II) dithiocarbamates revealed that Pb···S TeB is the main structure-directing interaction of these solid structures. We analyzed these TeB-based contacts in Section 2.1, whereupon in Section 2.2 and Section 2.3 we attempted to verify how another type of σ-hole interaction, namely HaB, affects the structural organization of the dithiocarbamates.

For this work, we used a combination of the lead(II) complex [Pb(S_2_CNEt_2_)_2_] with strong (and potentially tetrafunctional) HaB donors, such as tetraiodoethylene (C_2_I_4_). The co-crystallization of [Pb(S_2_CNEt_2_)_2_] with C_2_I_4_ gave the co-crystal, [Pb(S_2_CNEt_2_)_2_]∙½C_2_I_4_, in which, as we observed, the supramolecular organization of the X-ray solid-state structure is largely determined by an interplay between Pb⋯S TeB and I⋯S HaB. We found that despite a structure-directing contribution of HaB in the structure of [Pb(S_2_CNEt_2_)_2_]∙½C_2_I_4_, the TeBs from the parent complex, [Pb(S_2_CNEt_2_)_2_], remain unchanged and the co-crystallization with the HaB donor provides only a small change in the crystal parameters. All our findings are consistently detailed in the following sections.

## 2. Results

### 2.1. CSD Search: Structural Features of Homoleptic Lead(II) Dithiocarbamates

The supramolecular structure of solid lead(II) dithiocarbamates is determined by the ability of lead(II) sites to form TeB(s). The CSD search revealed 19 structures of [Pb(S_2_CNRR’)_2_], including 15 structures (R*_f_* ≤ 5.3%) exhibiting monomeric, oligomeric, and polymeric supramolecular motifs (Figure 1). Some complexes of the type [Pb(S_2_CNRR’)_2_] (RR’ = CH_2_Ph/CH_2_CH_2_(thienyl-2) AGABOB, (CH_2_Ph)_2_ QECCAE, and QECCAE01) are monomeric (**A**) without noticeable Pb-involving contacts; in these cases, intermolecular Pb–S distances exceed 4.10 Å (higher than the Bondi radii [68] sum, Σ_vdw_(Pb + S) = 3.82 Å).

We also identified two types of oligomeric structures. The structure of [Pb{S_2_CN(Me)CH_2_Ph}_2_] (YEDQII) represents a TeB-based tetramer (Figure 1B), in which each of two central molecules are involved in four Pb⋯S TeBs with two neighboring dithiocarbamate ligands. Another two peripheric molecules are involved in three contacts, each one with a neighboring molecule. A particular case is the structure of [Pb(S_2_CN*^n^*Pr_2_)_2_] (JADJIH), whose supramolecular aggregate consists of the molecular dimer, [Pb(S_2_CN*^n^*Pr_2_)_2_]_2_, which exhibits inter- and intramolecular TeBs, and the two mononuclear [Pb(S_2_CN*^n^*Pr_2_)_2_] entities linked to the molecular dimer via four TeBs (Figure 1C).

In the structure of [Pb(S_2_CNCy_2_)_2_] (BEQWUQ), each metal atom forms only one Pb⋯S TeB to give a 1D chain. However, if the other Pb⋯S long contact (3.94 Å) (which exceeds Bondi Σ_vdw_ 3.82 Å) is taken into account, this pattern might be attributed to chains, where each of the two neighboring molecules forms two mutual Pb···S contacts (Figure 1D). Another architecture of 1D chains (Figure 1E) was observed for [Pb(S_2_CNRR’)_2_] (RR’ = *^i^*Pr_2_ IPTCPB01, Et/*^i^*Pr NAYNUW and NAYNUW01, Et/Cy XAVYAU), where each complex forms two mutual contacts with the neighboring molecule, thus functioning as a TeB donor in one case and as a TeB acceptor in the other case.

The crystal structures of [Pb(S_2_CNRR’)_2_] (RR’ = Me/CH_2_Ph HABGAU, (CH_2_)_5_ JORRIU, Me_2_ MTCBPB, and Et_2_ PBETCA02) display 1D head-to-tail chains (Figure 1F). The 1D chains in the infinite polymeric structure of [Pb{S_2_CN(CH_2_)_4_}_2_] (NINDUJ) are based on pentafurcated intermolecular contacts; each contact includes four Pb···S and one Pb⋯Pb interaction (Figure 1G).

It is clear from the performed CSD search that the supramolecular organization of the crystal structures of lead(II) dithiocarbamates is greatly determined by Pb···S TeBs. These TeBs provide an assembly furnishing either tetrameric or polymeric structures. In some instances, Pb···S contacts are not formed, and the crystal structure motifs are determined by other non-covalent contacts, such as, for instance, H···S hydrogen bonds.

In the context of this study, we analyzed the structure of the parent complex [Pb(S_2_CNEt_2_)_2_] (PBETCA02, **1**) employed for the co-crystallization (Section 2.2). This structure belongs to type F (Figure 1), where the complex forms 1D infinite chains determined by the Pb···S TeB. The coordination environment of the lead(II) ion exhibits 4-coordinated distorted pseudotrigonal pyramidal geometry. The Pb–S distances range from 2.7301(17) to 2.9050(13) Å, and the bite angles ∠S–Pb–S are 63.41(3) and 64.68(4)°; the interligand ∠S–Pb–S angles are in the range of 82.92(4)–135.90(5)°. The neighboring complexes are linked via Pb⋯S TeB [4] (Figure 2): each complex behaves as TeB donor toward two S atoms from a neighboring molecule (either from one or from different ligands), providing two σ-holes at the Pb^II^ center for two Pb···S contacts, and as a TeB acceptor toward another neighboring **1**, thereby providing two σ-hole-accepting S atoms.

### 2.2. Crystallizations and Structural Motifs of the XRD Structures

As a next step of our study, we performed the crystallization of **1** with different HaB donors: 1,2-diiodotetrafluorobenzene, 1,4-diiodotetrafluorobenzene, 1,3,5-triiodotriafluorobenzene, and tetraiodoethylene (C_2_I_4_). However, only in the case of C_2_I_4_, we obtained crystals suitable for XRD, namely **1**∙½C_2_I_4_. This co-crystal was then studied by single-crystal XRD (see Section 4.2 and Section 4.3 for details). The structure of **1**∙½C_2_I_4_ is composed by one type of **1** and one type of C_2_I_4_. Complex **1** exhibits a 4-coordinated distorted pseudotrigonal pyramidal geometry, which is rather typical for lead(II) dithiocarbamates [69,70]. The Pb–S distances are in the range of 2.7312(9)–3.0200(11) Å, while the bite angles ∠S–Pb–S are 62.04(3) and 66.10(3)°, and the interligand ∠S–Pb–S angles lie in a broad interval spanning from 81.89(3) to 140.23(3)°. The neighboring complexes are linked to each other via mutual Pb⋯S TeB [4] (Figure 3).

Each metal center is involved in two TeBs with two different S sites (Appendix A), thus accomplishing an infinite double zig-zag chain (Figure 4). Considering these TeBs, the coordination environment of each lead(II) is thus completed to a distorted octahedral. The S2 and S1 atoms are linked to C_2_I_4_ via HaB (Figure 5, Appendix A). All four iodine atoms of C_2_I_4_ form HaBs and these contacts join TeB-based zig-zag chains to give a 3D structure.

The search of CSD revealed only one structure of **1** with *R*_w_ < 5% (CSD refcode: PBETCA02; *R*_w_ = 1.75%). In PBETCA02, similar to the structure of **1**∙½C_2_I_4_, the lead(II) site exhibits a 4-coordinated distorted pseudotrigonal pyramidal geometry. The other geometric parameters are also very similar: the Pb–S distances are in the range of 2.7301(17)–2.9050(13) Å, the bite angles ∠S–Pb–S are 63.41(3) and 64.68(4)°, and the interligand ∠S–Pb–S angles span from 82.92(4) to 135.90(5)°. Each metal center forms two Pb⋯S TeBs with the neighboring **1**. Although the 1D chains are the main structural motives of both **1** and **1**∙½C_2_I_4_, their architectures are different (Figure 6).

We performed the Hirshfeld surface analysis [71] for the XRD structures of **1** (PBETCA02) and co-crystal **1**∙½C_2_I_4_ to verify what kind of intermolecular contacts provide the largest contributions to the crystal packing for both structures. For the visualization, we used a mapping of the normalized contact distance (*d*_norm_); its negative value enables the identification of molecular regions (red circle areas) of substantial importance for the recognition of short contacts (Figure 7). For both structures, the shortest contacts are represented by Pb⋯S TeBs, while **1**∙½C_2_I_4_ I⋯S HaB contacts are also clearly visible, and they provide a substantial contribution.

### 2.3. Theoretical Calculations

The calculated (PBE [72] -D3 [73]/def2-TZVP [74,75]) electrostatic surface (ρ = 0.001 e/bohr^3^) [76] potentials [77,78,79] for **1** and C_2_I_4_ are based on the experimentally obtained coordinates and these potentials are positive for the Pb^II^ site and all I atoms (Figure 8), which exhibit σ-holes; the maximum σ-hole potential on the iodine atoms of C_2_I_4_ is larger than that on the Pb atom in **1** (27.0 vs. 9.8–13.4 kcal/mol).

To get a deeper insight into the nature of non-covalent interactions in co-crystal **1**∙½C_2_I_4_, we additionally performed two types of calculations. To model the whole system, the calculations under periodic boundary conditions (*crystal* model) were carried out in CP2K [80,81,82,83,84,85,86] using a PBE [72]-D3 [87] functional and jorge-DZP-DKH [88,89,90,91] full-electron basis set with the Douglas–Kroll–Hess 2nd-order scalar relativistic calculations requesting relativistic core Hamiltonian [92,93]. Concurrently, for a more detailed analysis, the *cluster* models were calculated in Gaussian-09 [94] using the same PBE [72]-D3 [73] functional and def2-TZVP [74,75] basis set, which contain two molecules of **1** only (two types of clusters accordingly to these TeBs) or the system **1** plus C_2_I_4_ (also two types of clusters). Both *crystal* and cluster models were based on the experimentally determined coordinates. For a more detailed description, see Computational (Section 4.4).

To get a deeper insight into the nature of non-covalent interactions in co-crystal **1**∙½C_2_I_4_, we additionally performed two types of calculations. To model the whole system, the calculations under periodic boundary conditions (*crystal* model) were carried out in CP2K [80,81,82,83,84,85,86] using a PBE [72]-D3 [87] functional and jorge-DZP-DKH [88,89,90,91] full-electron basis set with the Douglas–Kroll–Hess 2nd-order scalar relativistic calculations requesting relativistic core Hamiltonian [92,93]. Concurrently, for a more detailed analysis, the *cluster* models were calculated in Gaussian09 [94] using the same PBE [72]-D3 [73] functional and def2-TZVP [75,75] basis set, which contain two molecules of **1** only (two types of clusters accordingly to these TeBs) or the system **1** plus C_2_I_4_ (also two types of clusters). Both *crystal* and cluster models were based on the experimentally determined coordinates. For a more detailed description, see Computational (Section 4.4).

The QTAIM analysis [95,96,97] performed for the crystal and cluster models demonstrated the presence of bond critical points (3, −1) (abbreviated as BCP) between the I and S atoms as well as BCPs corresponding to the Pb⋯S TeB interactions (Table 1). Consideration of the negative and small values of the BCP sign(λ_2_)ρ(**r**) values indicated the attractive and non-covalent nature of the Pb⋯S and I···S interactions [98]. The conclusion on the non-covalent nature is based on their close to zero positive energy density (0.000–0.001 hartrees/bohr^3^) and the balance of the Lagrangian kinetic energy G(**r**) and the potential energy density V(**r**) (−G(**r**)/V(**r**) ≥ 1) on the corresponding BCPs [97]. The same topological parameters (Appendix A, the ESI) were found for the Pb⋯S interactions in crystal and cluster models of **1** and [Pb(S_2_CN*^n^*Pr_2_)_2_] of the CSD PBETCA02 and IPTCPB01 structures, respectively. According to the performed NCI analysis, the surfaces of the reduced density gradient (RDG) [98] with a 0.35 e^−⅓^ value with assigned negative sign(λ_2_)ρ(**r**) values on them surround both BCPs of the I⋯S (Figure 9) and Pb⋯S (Figure 10) interactions. Hence, the existence of the non-covalent interactions was confirmed by NCI analysis.

Wiberg bond indexes [99,100,101], which were calculated for the I⋯S HaBs and Pb⋯S TeBs in the cluster model in the natural atomic partitioning scheme [102,103], are within the 0.08–0.14 range (Table 1). These values are lower than the typical values for coordinative bonds [104,105], but still show some covalent contributions, which are especially noticeable for the S3–Pb1⋯S1 interactions.

The charge transfer from **1** to C_2_I_4_ was evaluated by calculation of the sums of the natural population analysis (NPA) [102,103] atomic charges in the model clusters (**1**)∙(C_2_I_4_) (Figure 9), which correspond to two different interactions, where the C_2_I_4_ sums are −0.069 and −0.084 e for the C1S–I1S⋯S1 and C1S–I2S⋯S2 interactions, respectively.

The electron localization function (ELF) [106,107,108] projections (useful for the location of shared and lone pair areas), along with QTAIM bond critical points and paths [65,67,109,110,111] can be employed to reveal electron donating/accepting roles of atoms involved in non-covalent interactions. The projections for the I⋯S HaBs carried out in the crystal and cluster models (Figure 11) show that the I⋯S bond paths go through S lone pair areas (with high ELF values; orange zones) and also between I lone pair areas (with low ELF areas; green zones). Thus, the S atoms are electron donating partners toward iodine σ-holes. Correspondingly, the I⋯S interactions can be treated as HaBs, according to the IUPAC recommendations for the identification of HaB [33].

The same projections (Figure 12) obtained for the Pb⋯S interactions also demonstrate the Pb⋯S bond paths lie on the S lone pair areas and far from the Pb lone pair areas. Since TeBs in many respects are analogous to HaBs [112] and Pb atoms behave as an electron acceptor toward sulfur, the Pb⋯S interactions can be attributed to TeBs.

The strength of the I⋯S and Pb⋯S interactions were calculated as a difference between the energy of a cluster and a sum of the monomer’s energies, thereby taking into account the basis set superposition error (BSSE) [113] using a counterpoise procedure. The energies (Table 2) were calculated for the model clusters (**1**)∙(C_2_I_4_) with the C–I⋯S interactions or for the (**1**)_2_ clusters with the S–Pb⋯S TeBs. Notably, the TeB energies are comparable with those calculated for the model (**1**)_2_ (−6.90 kcal/mol) and ([Pb(S_2_CN*^n^*Pr_2_)_2_])_2_ (−5.48 kcal/mol) clusters (Appendix A, the ESI) of the CSD PBETCA02 and IPTCPB01 structures, respectively. Notably, the geometrical parameters of non-covalent contacts and energies of HaB and TeB demonstrate a certain relationship. Thus, the normalized contact distances for Pb···S and I···S are comparable and fall into the 0.85–0.88 range, which corresponds to the close dimerization energies spanning from −5.41 to −11.37 kcal/mol.

In summary, the results of the performed theoretical calculations confirmed the existence and the non-covalent nature of the I⋯S HaB and Pb⋯S TeB interactions for the crystal and cluster models of **1**∙½C_2_I_4_. We also revealed the electron donating/accepting roles of the interacting atoms from the ELF and NPA calculations. This attribution is in agreement with the experimental angle parameters of the corresponding interactions. The energies of the I⋯S HaB and Pb⋯S TeB interactions span the range from −5.41 to −11.37 kcal/mol.

## 3. Discussion

The results of this study can be considered from at least two perspectives. In a narrow sense, we established that a common feature of the structure of **1**∙½C_2_I_4_ is the coexistence and interplay of Pb⋯S TeBs and I⋯S HaBs structure-directing interactions. The incorporation of the HaB donor in the crystal structure of **1** to give **1**∙½C_2_I_4_ provides, only slightly, an **F**-to-**E** type (Figure 6) change of the parent TeB-based supramolecular motifs. Regardless of the structural changes that occurred during the co-crystallization, the structure of **1**∙½C_2_I_4_ preserves the Pb⋯S TeB interactions. The number of Pb⋯S contacts involving each molecule of **1** and the coordination geometry of the complex remain intact, while the interaction with the HaB donor gives an additional supramolecular motif (Figure 13). As can be inferred from the examination of our results, the TeB bond determines both the structure of the homoleptic lead dithiocarbamate and the co-crystal, and the effects of TeB and HaB are comparable. An analysis of the theoretical calculation data, which were performed for the crystal and cluster models of **1**∙½C_2_I_4_ and confirms the non-covalent nature of both Pb⋯S TeB (−5.41 and −7.78 kcal/mol) and I⋯S HaB (−7.26 and −11.37 kcal/mol) interactions, as well as indicating that these non-covalent forces are comparable in their energies—while very different in regard to their physical natures.

The results of this work are consistent with those recently reported [66]. In the latter work, we found that the co-crystallization of [Ni(S_2_COEt)_2_] with 1,4-diiodotetrafluorobenzene (1,4-FIB) and 1,3,5-triiodotrifluorobenzene (1,3,5-FIB)—to give the co-crystals [Ni(S_2_COEt)_2_]·2(1,4-FIB) and [Ni(S_2_COEt)_2_]·2(1,3,5-FIB)—leads to a change in the geometry of the Ni···S contacts, but not to their disappearance. In both cases—adducts of nickel(II) dithiocarbonate and lead(II) dithiocarbamate—correlate with the high thiophilicity of these two metals, which is indirectly manifested in the pronounced ability of nickel and lead to form sulfides.

In a broader sense, our experimental observations and obtained computational data agree well with our recent studies that focused on the various approaches of HaB-involving supramolecular assembly of transition metal dithiocarbamates and -carbonates. These works [65,66,67] utilized square-planar late transition metal(II) complexes, which can act as *d*_z_^2^-orbital nucleophilic components of non-covalent interactions (Figure 14).

Although the non-transition metal(II) ion in [Pb(S_2_CNEt_2_)_2_] forms four coordination bonds and exhibits a 2+ oxidation state, it is different from the square-planar late transition metal(II) dithiocarbamates. The choice of another metal center, namely Pb^II^ instead of the *d*^8^-metals, affects the types of non-covalent interactions involved in the assembly: a positively-charged lead(II) center behaves as a σ-hole donor and functions as a component of TeB.

From the viewpoint of NCI-based assembly, [Pb(S_2_CNEt_2_)_2_] bears four σ-hole-accepting centers at the S-atoms of the two dithiocarbamate ligands, and in this regard, the lead(II) site is similar to the platinum group metal(II) dithiocarbamates (Figure 14). At the same time, the distinction between the geometries of [Pb(S_2_CNEt_2_)_2_] and [M(S_2_CNEt_2_)_2_] (M = Ni, Pd, Pt) can result in different HaBs distribution, thus providing different supramolecular assembly patterns. The consideration of the structures of the transition and non-transition complexes demonstrated how the identity of metal centers affects the geometry and composition on NCI-based supramolecular assembly of dithiocarbamate or -carbonate complexes.

## 4. Materials and Methods

### 4.1. Materials and Instrumentation

Complex **1** was prepared according to the published procedure [69]. Other reagents and solvents were obtained from commercial sources and used as received. The Cambridge Structural Database search was performed by using ConQuest software [114] (version 2022.1.0; access date: 15 March 2022).

### 4.2. Co-Crystallization

A mixture of [Pb(S_2_CNEt_2_)_2_] (10.0 mg, 0.020 mmol) and C_2_I_4_ (5.3 mg, 0.010 mmol) was dissolved in dichloromethane (3 mL) under ultrasonic treatment and filtered off through a PTFE syringe filter (0.45 μm) from minor, solid impurities. The reaction mixture was then left to stand at room temperature for slow evaporation. Pale yellow crystals of **1**∙½C_2_I_4_, suitable for XRD, were obtained after 4–5 days.

### 4.3. X-ray Structure Determination

Suitable single-crystals of **1**∙½C_2_I_4_ were fixed on micro-mounts, placed on an Xcalibur, Eos diffractometer (monochromated Mo Kα radiation, λ = 0.71073), and measured at 100(2) K using. Using Olex2 [115], the structure was solved with the ShelXT [116] (structure solution program using Intrinsic Phasing) and refined with the ShelXL [117] refinement package using Least Squares minimization. Supplementary crystallographic data for this paper have been deposited at the Cambridge Crystallographic Data Centre (CCDC 2205815) and can be obtained free of charge via www.ccdc.cam.ac.uk/data_request/cif, accessed on 7 September 2022. The table with the crystal data and structure refinement for **1**∙½C_2_I_4_ can be found in the ESI.

### 4.4. Computational Details

Single-point DFT calculations based on experimentally determined coordinates with periodic boundary conditions using the Gaussian/augmented plane wave (GAPW) [118] basis set with a 350 Ry and a 50 Ry relative plane-wave cut-offs for the auxiliary grid, the PBE [72]-D3 [73,87] functional, and full-electron jorge-DZP-DKH [88,89,90,91] basis for *crystal* (1 × 1 × 1 cell) models of **1**∙½C_2_I_4_, **1** (structure PBETCA02), and [Pb(S_2_CN*^i^*Pr_2_)_2_] (structure IPTCPB01) were performed in the CP2K-8.1 program [80,81,82,83,84,85,86] with the Douglas–Kroll–Hess 2nd-order scalar relativistic calculations requested relativistic core Hamiltonian [92,93]. The 0.500 r_loc_ parameter was applied for Pb atoms. The 1.0 × 10^−6^ Hartree convergence was achieved for the self-consistent field cycle in the Γ-point approximation. The analogous methodology was previously used for the investigation of the related halogen-bonded systems [119]. The gas-phase study for cluster models was performed in the same PBE-D3 level of theory in Gaussian-09 [94] with the def2-TZVP [74,75] basis set. The basis set superposition error (BSSE) for the calculation of interaction energies has been corrected using the counterpoise method [113]. Electron localization function (ELF) [106,107,108] projection analysis and Bader [95,96,97] Atoms-In-Molecules topological analysis of electron density (QTAIM) were performed and visualized in Multiwfn 3.8 [120]. A non-covalent interactions (NCI) [98] analysis of the reduced density gradient (RDG) as well as the analysis of the electrostatic surface (ρ = 0.001 e/bohr^3^) [76] potentials [77,78,79] (ESP) were carried out in Multiwfn 3.8 and visualized in VMD 1.9.3 [121]. Wiberg bond indexes (WBI) [99,100,101] in natural atomic partitioning scheme and natural population analysis (NPA) [102,103] atomic charges were calculated for cluster models using GENNBO utility in NBO 7.0 [122] based on 0.47 files generated in Multiwfn 3.8. At the preliminary stage of our work, we conducted several calculations using various functionals. An inspection of our results indicates that the change of functional does not affect the results and it is clear that PBE-D3 is a suitable functional, which is conventionally used in many studies of non-covalent interactions [119,123]

### 4.5. Details of the Hirshfeld Surface Analysis

HSA was carried out using the CrystalExplorer program [71,124,125]. The contact distances (*d_norm_*) [126], based on Bondi vdW radii [68,127], were mapped on the Hirshfeld surface (Appendix A). In the color scale, the negative values of *d_norm_* were visualized by red color, thereby indicating the contacts that were shorter than Σ*R*_vdW_. The values represented in white denote the intermolecular distances that are close to the vdW contacts with *d_norm_* equal to zero. In turn, the contacts longer than Σ*R*_vdW_ with positive *d_norm_* values were colored in blue.

## 5. Conclusions

We found that the co-crystallization of [Pb(S_2_CNEt_2_)_2_] with C_2_I_4_ gave the co-crystal, [Pb(S_2_CNEt_2_)_2_]∙½C_2_I_4_, in which the supramolecular organization of the X-ray solid-state structure is largely determined by an interplay between Pb⋯S TeB and I⋯S HaB. Despite a structure-directing contribution of HaB in the structure of the co-crystal, the TeBs from the parent complex, [Pb(S_2_CNEt_2_)_2_], remain unchanged and the co-crystallization with the HaB donor provides only a small change of the crystal parameters. An analysis of the theoretical calculation data, performed for the crystal and cluster models of [Pb(S_2_CNEt_2_)_2_]∙½C_2_I_4_, revealed the non-covalent nature of the Pb⋯S TeB (−5.41 and −7.78 kcal/mol) and I⋯S HaB (−7.26 and −11.37 kcal/mol) interactions and indicated that in the co-crystal, these non-covalent forces are similar in energy. Our experimental observations and appropriate computational data agree well with those reported in our recent studies that focused on the various approaches of HaB-involving supramolecular assembly of square-planar late transition metal dithiocarbamates and -carbonates [65,66,67]. The consideration of the structures of the transition and non-transition complexes demonstrated how the identity of metal centers affects the geometry and composition of NCI-based supramolecular assembly of dithiocarbamate or -carbonate complexes.

## Figures and Tables

**Figure 1 ijms-23-11870-f001:**
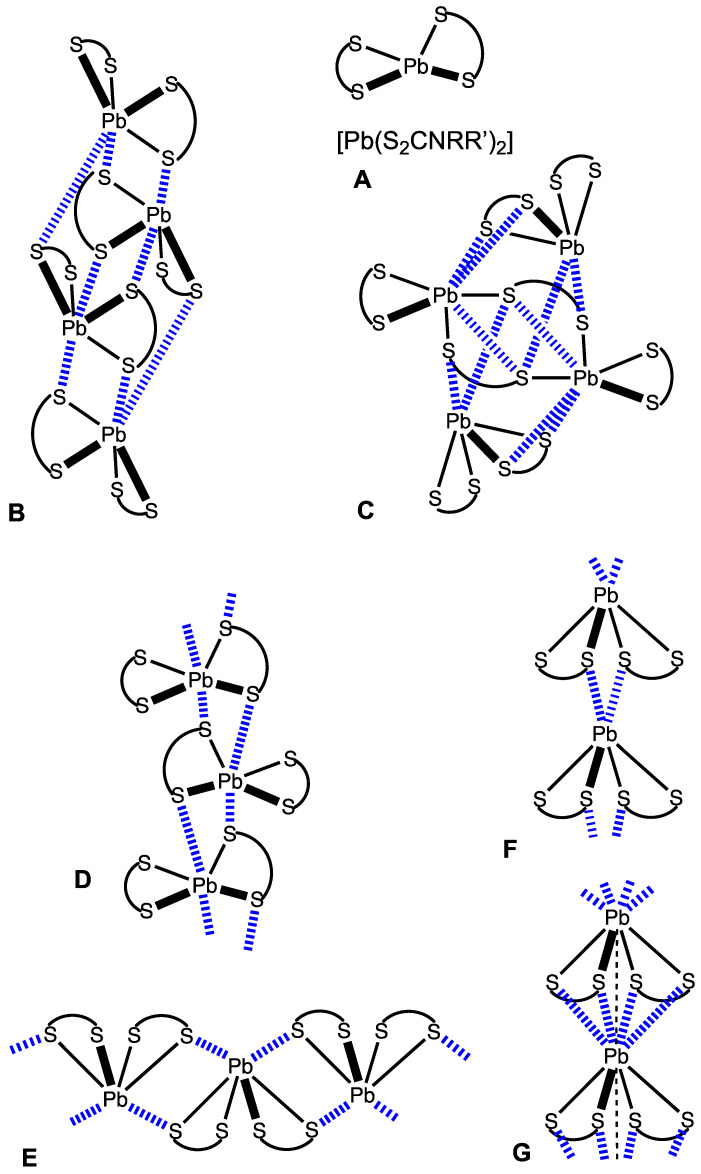
Supramolecular organization of lead(II) dithiocarbamates: monomeric (**A**) and two tetrameric (**B**,**C**) forms. Hereinafter, short M⋯S non-covalent contacts are shown by blue dotted lines, polymeric forms: two kind of chains, based on mutual Pb···S contacts (**D**,**E**), 1D head-to-tail chain (**F**), and 1D head-to-tail chain involving Pb⋯Pb interaction (**G**).

**Figure 2 ijms-23-11870-f002:**
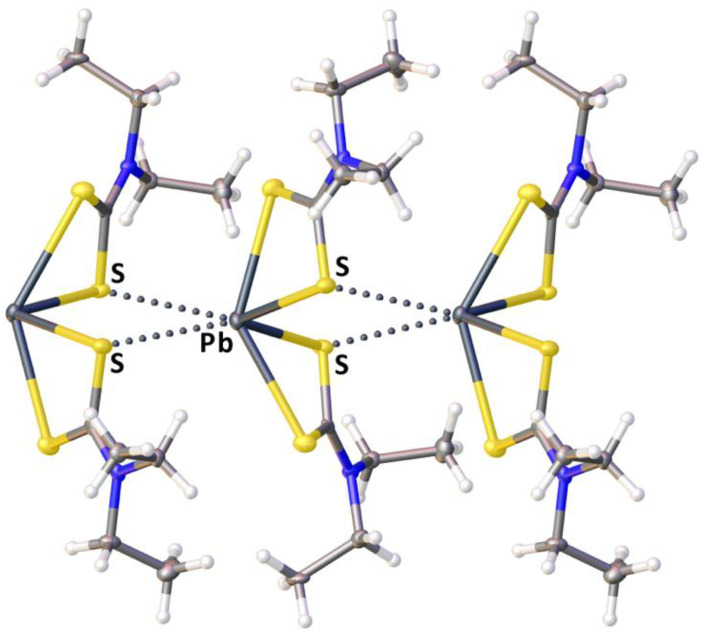
A fragment of the crystal packing of **1** (PBETCA02) showing Pb⋯S TeB contacts (dotted lines).

**Figure 3 ijms-23-11870-f003:**
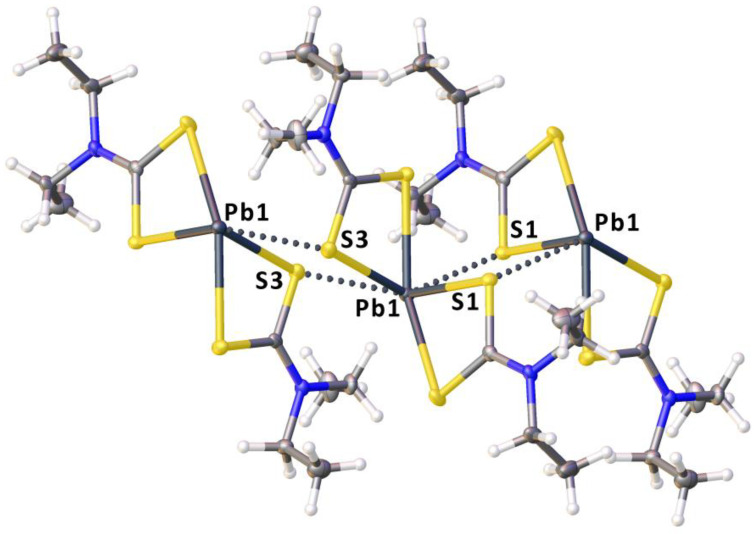
A fragment of the crystal packing of **1**∙½C_2_I_4_ showing Pb⋯S TeB contacts (dotted lines).

**Figure 4 ijms-23-11870-f004:**
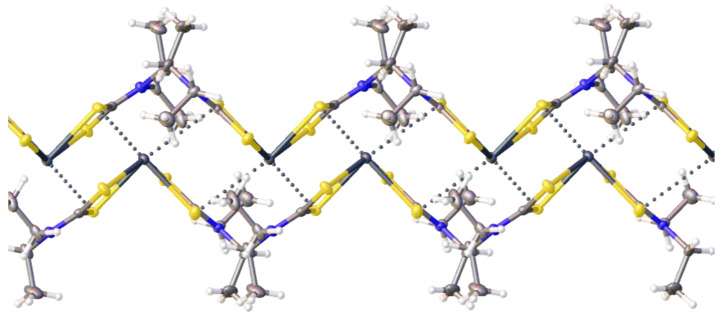
Intermolecular Pb⋯S TeB contacts-based double zig-zag chain in the crystal structure of **1**∙½C_2_I_4_.

**Figure 5 ijms-23-11870-f005:**
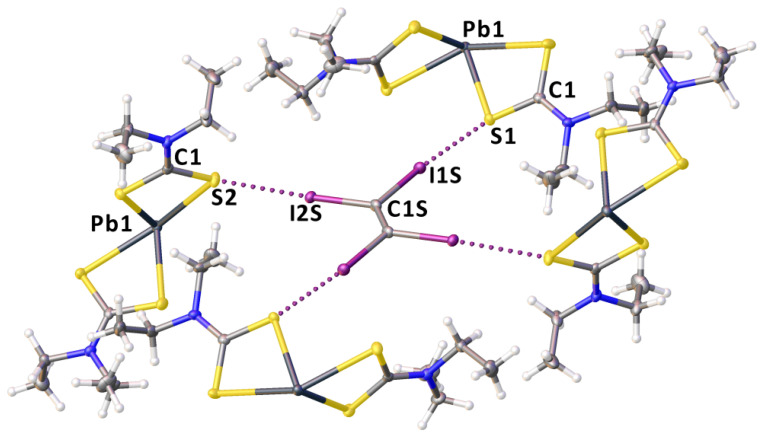
A fragment of crystal packing of **1**∙½C_2_I_4_ showing I⋯S HaB contacts (dotted lines).

**Figure 6 ijms-23-11870-f006:**
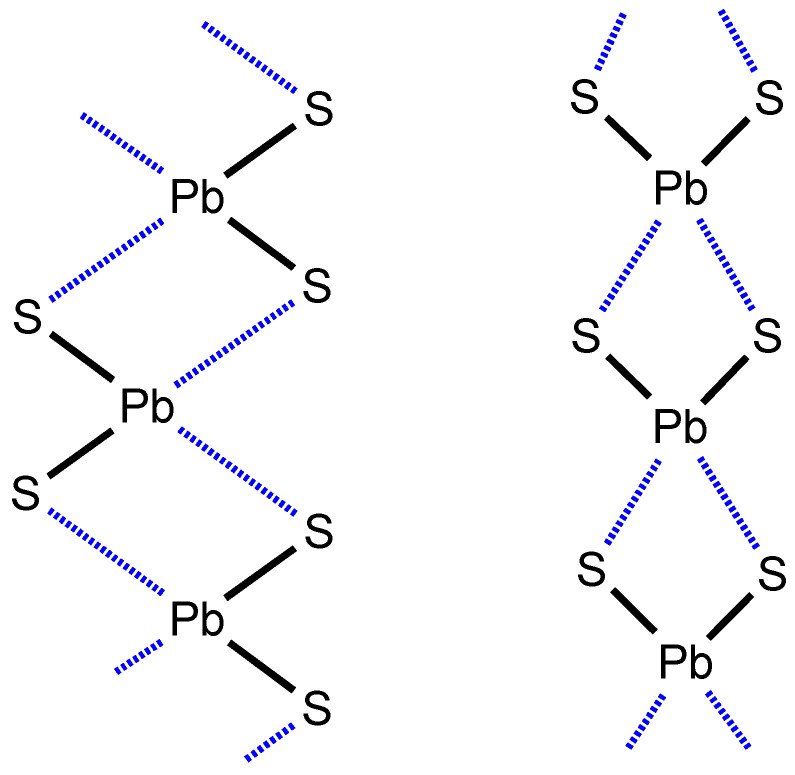
Pb···S-based supramolecular organization in the structures of **1** (PBETCA02, left panel) and **1**∙½C_2_I_4_ (right panel). Short M⋯S TeB contacts are shown by blue dotted lines.

**Figure 7 ijms-23-11870-f007:**
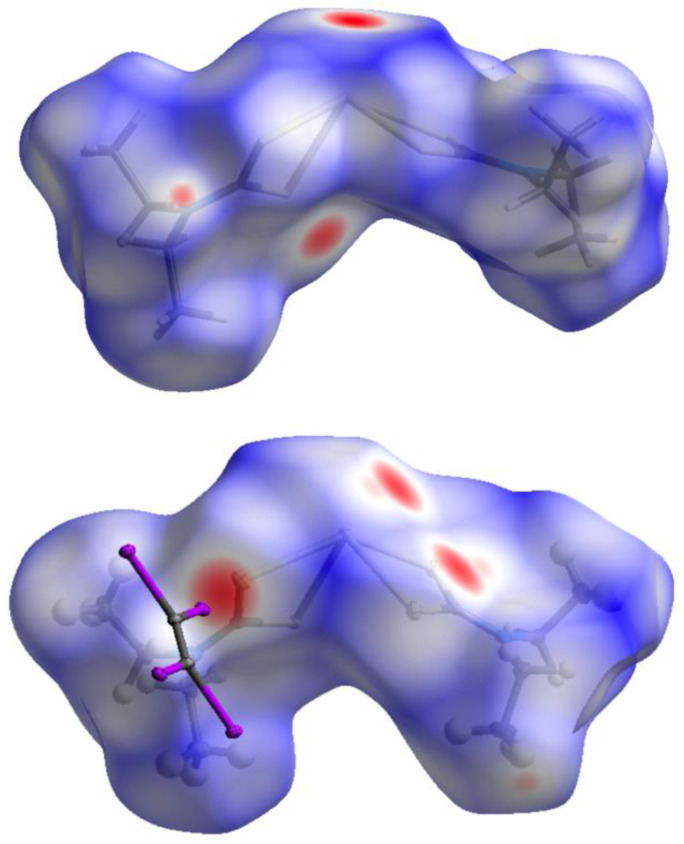
Visualization of the Hirshfeld surface, mapped over the normalized contact distance (*d*_norm_) of complex **1** in the structure of **1** (PBETCA02, top panel) and in co-crystal **1**∙½C_2_I_4_ (bottom panel).

**Figure 8 ijms-23-11870-f008:**
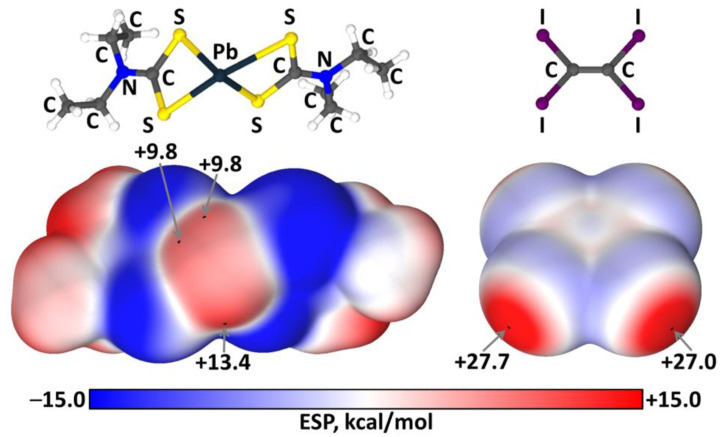
Electrostatic surface (ρ = 0.001 e/bohr^3^) potentials calculated for **1** (left) and C_2_I_4_ (right) molecules.

**Figure 9 ijms-23-11870-f009:**
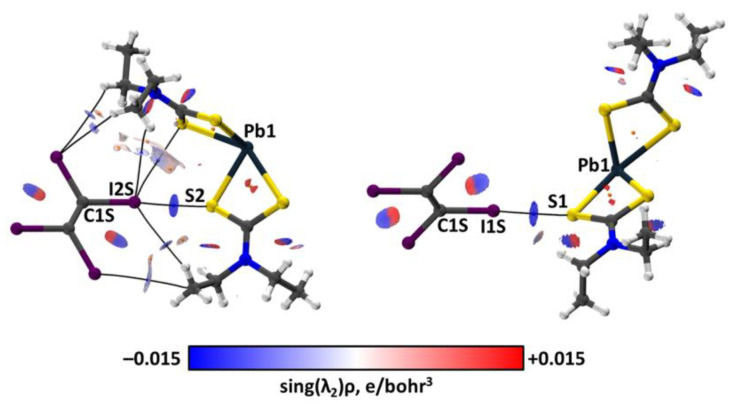
Visualization of QTAIM topological and NCI analyses for the model clusters (**1**)∙(C_2_I_4_) with two different I⋯S HaBs. Blue dots correspond to (3; −1) bond critical points, orange dots to (3; +1) ring critical points, and bond paths are shown as black lines. For non-covalent interactions, RDG = 0.35 e^−⅓^ half-transparent surfaces are colored from blue (sign(λ_2_)ρ(**r**) = −0.015 e/bohr^3^) to red (sign(λ_2_)ρ(r) = +0.015 e/bohr^3^) through white.

**Figure 10 ijms-23-11870-f010:**
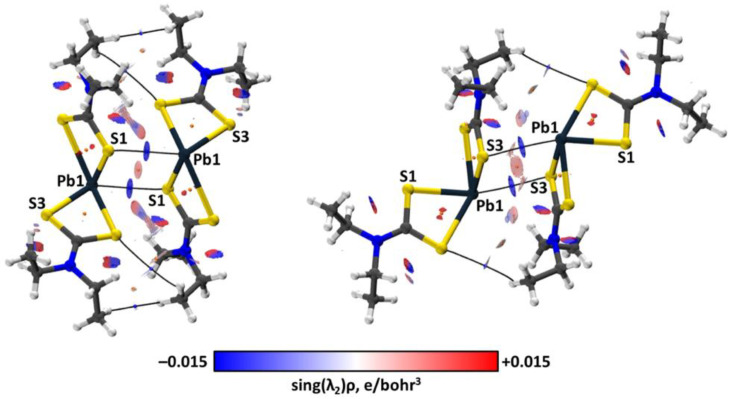
Visualization of QTAIM topological and NCI analyses for the model clusters (**1**)_2_ exhibiting two different Pb⋯S TeBs. Blue dots correspond to (3; −1) bond critical points, orange dots to (3; +1) ring critical points, and bond paths are shown as black lines. For non-covalent interactions, RDG = 0.35 e^−⅓^ half-transparent surfaces are colored from blue (sign(λ_2_)ρ(**r**) = −0.015 e/bohr^3^) to red (sign(λ_2_)ρ(**r**) = +0.015 e/bohr^3^) through white.

**Figure 11 ijms-23-11870-f011:**
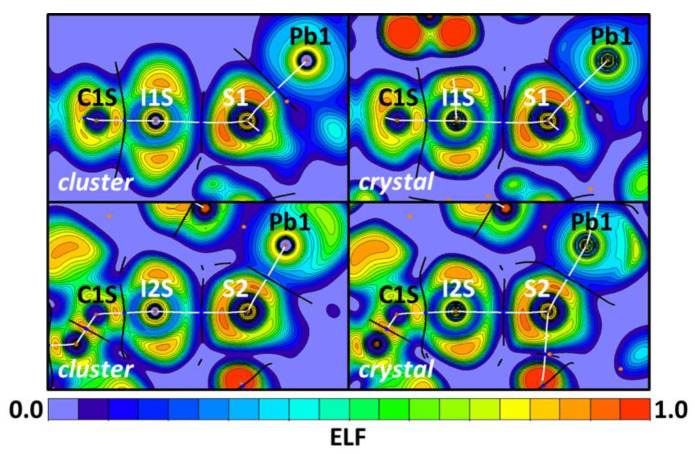
ELF projections through the Pb–S⋯I planes for the cluster (left) and crystal (right) models. QTAIM ρ(**r**) topological pale brown nuclear (3, −3), blue bond (3, −1), and orange ring (3, +1) critical points are drawn with white bond paths and black interatomic zero-flux paths.

**Figure 12 ijms-23-11870-f012:**
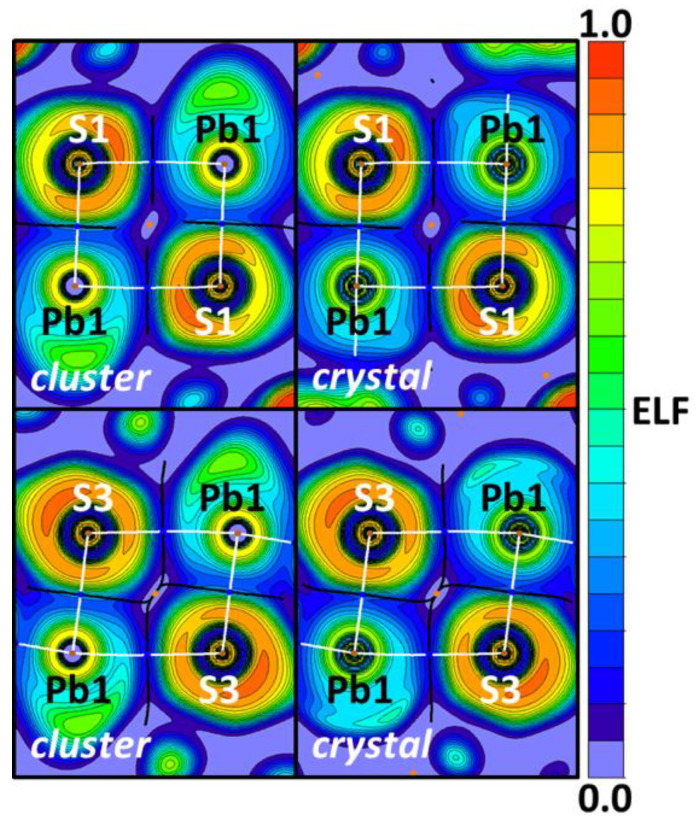
ELF projections through the Pb⋯S–Pb planes for the cluster (left) and crystal (right) models. QTAIM ρ(**r**) topological pale brown nuclear (3, −3), blue bond (3, −1), and orange ring (3, +1) critical points are drawn with white bond paths and black interatomic zero-flux paths.

**Figure 13 ijms-23-11870-f013:**
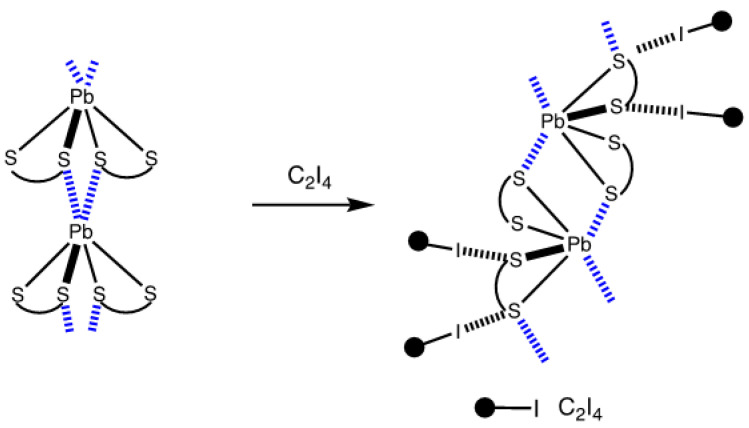
Changes in the supramolecular organization of **1** occurred by the incorporation of the HaB donor. The Pb⋯S contacts are shown by blue dotted lines, I⋯S HaBs are given as black dotted lines.

**Figure 14 ijms-23-11870-f014:**
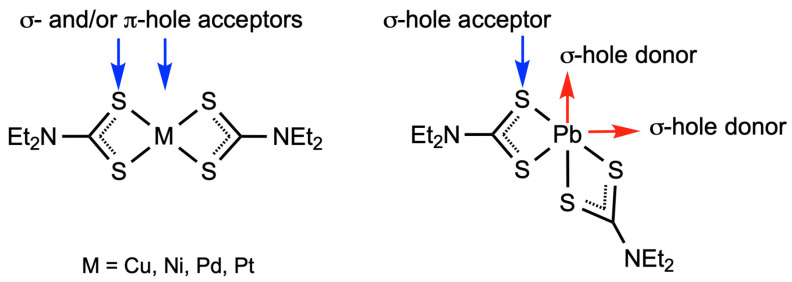
Comparison of [M(S_2_CNEt_2_)_2_] (M = Ni, Pd, Pt) and [Pb(S_2_CNEt_2_)_2_] as potential partners of NCIs.

**Table 1 ijms-23-11870-t001:** Parameters in (3, −1) bond critical points (the electron density with sign of λ_2_ sign(λ_2_)ρ(**r**) in e/bohr^3^, Laplacian of electron density ∇^2^ρ(**r**) in e/bohr^5^, the local electronic energy density H_b_, local electronic potential energy density V(**r**), and local electronic kinetic energy density G(**r**) in hartrees/bohr^3^) corresponding I⋯S HaBs and Pb⋯S TeBs in both crystal and cluster models as well as Wiberg bond indexes (WBI) calculated for the cluster model.

Bond	Model	sign(λ_2_)ρ(r)	▽^2^ρ(r)	G(r)	V(r)	Hb	WBI
C1S–I1S⋯S1	crystal	−0.016	0.043	0.010	−0.009	0.001	
	cluster	−0.016	0.038	0.009	−0.008	0.001	0.08
C1S–I2S⋯S2	crystal	−0.019	0.046	0.011	−0.011	0.000	
	cluster	−0.018	0.042	0.010	−0.009	0.001	0.10
S3–Pb1⋯S1	crystal	−0.019	0.039	0.010	−0.009	0.001	
	cluster	−0.019	0.038	0.010	−0.010	0.000	0.14
S1–Pb1⋯S3	crystal	−0.016	0.032	0.008	−0.007	0.001	
	cluster	−0.016	0.032	0.008	−0.008	0.000	0.09

**Table 2 ijms-23-11870-t002:** BSSE-corrected dimerization energies Δ*E* (kcal/mol) for the (**1**)∙(C_2_I_4_) and (**1**)_2_ heterodimers of two types.

Cluster Type	Bond	Δ*E*
(**1**)∙(C_2_I_4_)	I1S⋯S1	−11.37
	I2S⋯S2	−7.26
(**1**)_2_	Pb1⋯S1	−7.78 ^a^
	Pb1⋯S3	−5.41 ^a^

^a^—is Δ*E*/2 values since two Pb⋯S interactions dissociated from cluster to two monomers.

## Data Availability

Not applicable.

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
