# Peer review of "Structure-Directing Interplay between Tetrel and Halogen Bonding in Co-Crystal of Lead(II) Diethyldithiocarbamate with Tetraiodoethylene"

_ijms, 2022, doi:10.3390/ijms231911870_

Round 1
Reviewer 1 Report
The work by Kukushkin and Bokach is a perfect research work in the field of non-covalent interactions with focus on tetrel and halogen bonding. They have big name in the field and they have checked different systems including organic, metal-organic and organometallic systems.
The combination of X-ray with complementary theoretical calculations to confirm the observed interactions in X-ray is the merit and high credit of their works. They are pioneer in halogen bonding between organic halogen sigma-donor system with metal complexes in which the organic halogen bond donor interact with both metal-bound halogen and metal center with high electron density such as Pt and Pd.
I keep tracking their publications and I always enjoy reading their papers. They have real expert team in the field with good facility in hand to pursue such research work.
I am quite familiar to their work and as a structural and computational chemist believe that they are perfect and they know what they are doing.
Therefore, I would like to accept this manuscript in the present form, AS IT IS.
Author Response
No corrections required. We thank the reviewer for positive assessment of our study.
Reviewer 2 Report
Authors reported the crystal structure of the cocrystal [Pb(S2CNEt2)2]∙½C2I4 obtained in the reaction lead(II) complex [Pb(S2CNEt2)2] with tetraiodoethylene (C2I4). The theoretical calculation data for Pb∙∙∙S tetrel bonding (TeB) and I∙∙∙S halogen bonding (HaB) were compared. Analysis confirmed noncovalent nature of these interactions and indicated that these forces are comparable in their energies. The number of cocrystals investigated (only one) leaves a certain degree of insufficiency. However, authors reviewed the CSD base in the field of Pb(II) dithiocarbamates, which enriched the discussion.
The DFT calculations were supported by Hirshfeld surface analysis, which should be discussed in the main manuscript in more details.
Moreover, the QTAIM results should be discussed more quantitatively to better characterize TeB and HaB interactions in [Pb(S2CNEt2)2]∙½C2I4. The relationship NCIs geometrical parameters (e.g. distances) form X-ray study (please include in ESI) and QTAIM topological and energetic parameters should be discussed.
The manuscript needs a minor editorial assistance.
The last good papers on the use of QTAIM method for the characterization of tetrel and other interactions in Pb(II) coordination polymers and organic compounds are worth to be cited (CrystEngComm 2021, 23, 6137–6162; Polyhedron 2022, 219, 115818).
Author Response
Reviewer 2
The DFT calculations were supported by Hirshfeld surface analysis, which should be discussed in the main manuscript in more details.
Our response. According to the reviewer suggestion we moved discussion relevant to the Hirshfeld surface analysis (along with appropriate figure) from the ESI to the main text.
The following text was transferred to the main body of the manuscript: “We performed the Hirshfeld surface analysis for the XRD structures of 1 (PBETCA02) and cocrystal 1∙½C2I4, to verify what kind of intermolecular contacts provides the largest contributions to the crystal packing for both structures. For the visualization, we used a mapping of the normalized contact distance (dnorm); its negative value enables the identification of molecular regions (red circle areas) of substantial importance for recognition of short contacts (Figure 8). For both structures, the shortest contacts are represented by Pb∙∙∙S TeBs, while for 1∙½C2I4 I∙∙∙S HaB contacts also clearly visible and they also provide a substantial contribution.
Figure 8. Visualization of the Hirshfeld surface, mapped over the normalized contact distance (dnorm), of complex 1 in the structure of 1 (PBETCA02, top panel) and in cocrystal 1∙½C2I4 (bottom panel, right).”
Moreover, the QTAIM results should be discussed more quantitatively to better characterize TeB and HaB interactions in [Pb(S2CNEt2)2]∙½C2I4. The relationship NCIs geometrical parameters (e.g. distances) form X-ray study (please include in ESI) and QTAIM topological and energetic parameters should be discussed.
Our response. We transferred Table 1 to the ESI. Energetic parameters of HaB and TeB were compared based on calculation of BSSE-corrected dimerization energies in corresponding dimers, see p. 10 of the revised manuscript. To emphasize the relationship between the NCIs geometrical and energetic parameters we added the following phrase: “Notably, geometrical parameters of noncovalent contacts and energies of HaB and TeB demonstrate certain relationship. Thus, the normalized contact distances for Pb···S and I···S are comparable and fall in the 0.85–0.88 range, that corresponds to the close dimerization energies spanning from –5.41 to –11.37 kcal/mol.”
The last good papers on the use of QTAIM method for the characterization of tetrel and other interactions in Pb(II) coordination polymers and organic compounds are worth to be cited (CrystEngComm 2021, 23, 6137–6162; Polyhedron 2022, 219, 115818).
Our response. We agree with these suggestions and we added the following recent references to the citation list (see refs. 21 and 44 in the revised manuscript):
[21] Kowalik, M.; Brzeski, J.; Gawrońska, M.; Kazimierczuk, K.; Makowski, M. Experimental and theoretical investigation of conformational states and noncovalent interactions in crystalline sulfonamides with a methoxyphenyl moiety. CrystEngComm 2021, 23, 6137-6162, doi:10.1039/D1CE00869B.
[47] Kowalik, M.; Masternak, J.; Brzeski, J.; Daszkiewicz, M.; Barszcz, B. Effect of a lone electron pair and tetrel interactions on the structure of Pb(II) CPs constructed from pyrimidine carboxylates and auxiliary inorganic ions. Polyhedron 2022, 219, 115818

Reviewer 3 Report
This study focus on the co-crystal which has X-ray structure that exhibits only a small change of the crystal parameters than those in the parent complex. Supramolecular organization of the co-crystal is largely determined by an interplay between Pb and I....S halogen bonding (HaB) with comparable contribution of these non-covalent contacts. The study is of great interest particularly for the experimental teams.
--The introduction should be improved and make them short.
--The author highly suggested to add single para regarding the limitation of the study, why this was conducted.
--The calculation should be repeated three time, just for the ease of reproducibility, particularly the theoretical section of energy calculation. Always, the theoretical study hold some uncertainty.
--The results and discussion section need flow and story while reading, some para make not-important sense to this study which can be omit.
-- The conclusion section should be revised for flow.
--The author highly suggested to add more recent article, i.e., 2022 to make the story unto date.
--The author should revise the manuscript for typo, grammatical error and for the flow.
Author Response
--The introduction should be improved and make them short.
Our response. One has to consider that that Int. J. Mol. Sci. belongs to the category of general chemistry journals. For this type of journals an extensive introduction with substantial number of references—which allows the unprepared reader to quickly get into the topic—is quite common. In addition, the two other reviewers were quite satisfied with the description given in introduction. Our suggestion is to leave it as is. However, if the reviewer insists on the reductions, we appreciate if he (or she) gives us more suggestions which concrete places should be omitted or moved to Results and Discussion.
--The author highly suggested to add single para regarding the limitation of the study, why this was conducted.
Our response. According to the reviewer suggestion we added the following phrase on p. 4: “As a next step of our study, we performed crystallization of 1 with different HaB donors: 1,2-diiodotetrafluorobenzene, 1,4-diiodotetrafluorobenzene, 1,3,5-triiodotriafluorobenzene, and tetraiodoethylene (C2I4). However, only in case of C2I4, we obtained crystals suitable for XRD, namely 1∙½C2I4. This cocrystal was then studied by single-crystal XRD (see sections 4.2–4.3 for details).”
--The calculation should be repeated three time, just for the ease of reproducibility, particularly the theoretical section of energy calculation. Always, the theoretical study hold some uncertainty.
Our response. At the preliminary stage of our work, we conducted several calculations using various functionals. Inspection of our results indicates that the change of functional does not affect the results and it is clear that PBE-D3 is a suitable functional, which is conventionally used in many studies of noncovalent interactions (see, for instance, relevant discussion and appropriate references in our recent reports: [122] Kinzhalov, M.A.; Ivanov, D.M.; Melekhova, A.A.; Bokach, N.A.; Gomila, R.M.; Frontera, A.; Kukushkin, V.Y. Inorg. Chem. Front. 2022, 9, 1655-1665, doi:10.1039/D2QI00034B; [126] I. S. Aliyarova, E. Yu. Tupikina, N. S. Soldatova, D. M. Ivanov, P. S. Postnikov, M. Yusubov, V. Yu. Kukushkin, Inorg. Chem., 2022, 61; 10.1021/acs.inorgchem.2c01858.).
We added this phrase and the references to the Computational methods section.
--The results and discussion section need flow and story while reading, some para make not-important sense to this study which can be omit.
Our response. We moved Table 1 to the ESI.
-- The conclusion section should be revised for flow.
Our response. We added the following brief Conclusion section at the end of the revised manuscript:
- Conclusion
We found, that the cocrystallization of [Pb(S2CNEt2)2] with C2I4 gave the cocrystal [Pb(S2CNEt2)2]∙½C2I4, in which, the supramolecular organization of the X-ray solid-state structure is largely determined by an interplay between Pb∙∙∙S TeB and I∙∙∙S HaB. Despite a structure-directing contribution of HaB in the structure of the cocrystal, the TeBs from the parent complex [Pb(S2CNEt2)2] remain unchanged and the cocrystallization with the HaB donor provides only a small change of the crystal parameters. Analysis of the theoretical calculation data, performed for the crystal and cluster models of [Pb(S2CNEt2)2]∙½C2I4, revealed the noncovalent nature of the Pb∙∙∙S TeB (–5.41 and –7.78 kcal/mol) and I∙∙∙S HaB (–7.26 and –11.37 kcal/mol) interactions and indicate that in the cocrystal these noncovalent forces are similar in energy. Our experimental observations and appropriate computational data agree well with those reported in our recent studies focused on various approaches to HaB-involving supramolecular assembly of square-planar late transition metal dithiocarbamates and -carbonates [2-4]. The consideration of the structures of the transition and non-transition complexes demonstrated how the identity of metal centers affects the geometry and composition on NCI-based supramolecular assembly of dithiocarbamate or -carbonate complexes.
--The author highly suggested to add more recent article, i.e., 2022 to make the story unto date.
Our response. We agree with the reviewer and we added four references of the year 2022 in the text. Namely, refs. 42–44, and 47–49 (p. 1):
[42] Khera, M.; Goel, N. Cooperative Effect of Noncovalent Interactions on Tetrel Bonding in Halogenated Silanes. ChemPhysChem 2022, 23, e202100776, doi:https://doi.org/10.1002/cphc.202100776
[43] Bhattarai, S.; Sutradhar, D.; Chandra, A.K. Strongly Bound π-Hole Tetrel Bonded Complexes between H2SiO and Substituted Pyridines. Influence of Substituents. ChemPhysChem 2022, 23, e202200146, doi:https://doi.org/10.1002/cphc.202200146
[44] Southern, S.A.; Nag, T.; Kumar, V.; Triglav, M.; Levin, K.; Bryce, D.L. NMR Response of the Tetrel Bond Donor. The Journal of Physical Chemistry C 2022, 126, 851-865, doi:10.1021/acs.jpcc.1c10121
[47] Kowalik, M.; Masternak, J.; Brzeski, J.; Daszkiewicz, M.; Barszcz, B. Effect of a lone electron pair and tetrel interactions on the structure of Pb(II) CPs constructed from pyrimidine carboxylates and auxiliary inorganic ions. Polyhedron 2022, 219, 115818, doi:https://doi.org/10.1016/j.poly.2022.115818
[48] Majumdar, D.; Frontera, A.; Gomila, R.M.; Das, S.; Bankura, K. Synthesis, spectroscopic findings and crystal engineering of Pb(ii)–Salen coordination polymers, and supramolecular architectures engineered by σ-hole/spodium/tetrel bonds: a combined experimental and theoretical investigation. RSC Advances 2022, 12, 6352-6363, doi:10.1039/D1RA09346K.
[49] Mahmoudi, G.; García-Santos, I.; Pittelkow, M.; Kamounah, F.S.; Zangrando, E.; Babashkina, M.G.; Frontera, A.; Safin, D.A. The tetrel bonding role in supramolecular aggregation of lead(II) acetate and a thiosemicarbazide derivative. Acta Crystallographica Section B 2022, 78, 685-694, doi:https://doi.org/10.1107/S2052520622005789.
--The author should revise the manuscript for typo, grammatical error and for the flow.
Our response. We did all our best to improve language of the manuscript and in some instances sought help of native English speakers. We hope that the current version of the manuscript, in this respect, is substantially improved than the starting one.

Round 2
Reviewer 2 Report
The revised manuscript can be accepted by the IJMS.
Reviewer 3 Report
The author addressed my concern up to some extend, I am willing to accept the manuscript in the current form.